Mass mortality of pearl oyster (Pinctada fucata (Gould)) in Japan in 2019 and 2020 is caused by an unidentified infectious agent

Matsuyama Tomomasa matsuym@fra.affrc.go.jp
Miwa Satoshi
Mekata Tohru
Matsuura Yuta
Takano Tomokazu
Nakayasu Chihaya
Fish Pathology Department, Aquaculture Research Department, National Research and Development Agency, Japan Fisheries Research and Education Agency , Minami-Ise, Mie Prefecture , Japan
Oehlmann Jörg
Electronic publication date: 2021 Sep 21
Publication date: 2021
Volume: 9
Electronic Location ID: e12180
Received 2021 Jun 3; Accepted 2021 Aug 29
Copyright: © 2021 Matsuyama et al.
Copyright year: 2021
Copyright holder: Matsuyama et al.
License: This is an open access article distributed under the terms of the Creative Commons Attribution License, which permits unrestricted use, distribution, reproduction and adaptation in any medium and for any purpose provided that it is properly attributed. For attribution, the original author(s), title, publication source (PeerJ) and either DOI or URL of the article must be cited.
License URL: https://creativecommons.org/licenses/by/4.0/

Keywords: Pearl oyster, Pinctada fucata, Mass mortality, Infectious disease, Virus, Atrophy, Shell disease

Funding: Food Safety and Consumer Affairs Bureau, the Ministry of Agriculture, Forestry, and Fisheries of Japan This study was supported by the grants from Food Safety and Consumer Affairs Bureau, the Ministry of Agriculture, Forestry, and Fisheries of Japan. The funders had no role in study design, data collection and analysis, decision to publish, or preparation of the manuscript.

==============================
Mass mortality of 0-year-old pearl oysters, Pinctada fucata (Gould), and anomalies in adults were observed in Japan’s major pearl farming areas in the summer of 2019 and 2020. Although adult oyster mortality was low, both adult and juvenile oysters underwent atrophy of the soft body, detachment of the mantle from nacre (the shiny inner surface of the valves), deposition of brownish material on the nacre, and loss of nacre luster. Infection trials were conducted to verify the involvement of pathogens in this phenomenon. Healthy adult pearl oysters were obtained from areas where this disease had not occurred to use as the recipients. The sources of infection were either affected adult oysters with atrophied soft bodies or batches of juveniles in which mortality had reached conspicuous levels. Transmission of the disease to the healthy oysters were tested either by cohabitation with affected oysters or by injections of the hemolymph of affected animals. The injection infection test examined the effects of filtration and chloroform exposure on the pathogen. Occurrence of the disease was confirmed by the appearance of brown deposits on the nacre and loss of nacre luster. The abnormalities of nacre were clearly reproduced in recipient shells in three out of four cohabitation trials with affected oysters. The disease was also reproduced in six out of six injection trails either with hemolymph filtered through 100 nm filter or with hemolymph treated with chloroform. In a serial passage with hemolymph injections, the disease was successfully transmitted through eight passages. These results suggest that the etiology of the disease is a non-enveloped virus with a diameter ≤100 nm.

Introduction

The Akoya pearl oyster, Pinctada fucata (Gould), is the major species that is used for pearl culture in Japan. In recent years, the annual production of pearls in Japan has been approximately 20 tons and the annual export value is approximately 30 billion yen, which is second only to scallops. More than 90% of Akoya pearls are produced in Ehime, Nagasaki, and Mie prefectures (Fig. 1).

Figure 1 Locations of major pearl farms and areas where healthy pearl oysters were obtained.

The prefectures painted in red and black indicate major pearl farming areas using P. fucata and areas where healthy pearl oysters were obtained, respectively.

Mass mortalities of juvenile Akoya oysters and adult Akoya oysters with atrophy of the soft body were reported on pearl oyster farms in Mie Prefecture in June 2019 and Ehime Prefecture in early August 2019. Later in 2019, similar cases were reported from major pearl production areas, such as Nagasaki, Kumamoto, and Oita prefectures. In particular, the death of 0-year-old oysters (hereinafter referred to as juveniles) was considerable. According to the inquiry conducted by the Mie Prefecture Fisheries Research Institute to multiple pearl farming companies in three regions of the prefecture, the mortalities of juveniles and adults were 56–74% and 23–24%, respectively in 2019, whereas those in typical years were 15% and 9–16%. In 2020, similar abnormalities occurred in all major pearl farming regions in the country, from June to November when the water temperature was high, attaining the peak in June or July.

To date, the major infectious diseases known to affect P. fucata oysters in Japan are Akoya oyster disease (Kurokawa et al., 1999; Matsuyama et al., 2017, 2018, 2019), black-spot shell disease (Sakatoku et al., 2018), marine birnavirus infection (Suzuki, Kamakura & Kusuda, 1998; Suzuki, Utsunomiya & Kusuda, 1998), and trematode infestation (Sakaguchi, 1968). However, the mortality or abnormalities that occurred in 2019 and 2020 differ from any of these diseases and are referred to herein as “summer atrophy”. In healthy pearl oysters, the outer edge of the mantle is in close contact with the edge of the nacre surface (Fig. 2D). In the oysters with summer atrophy, the mantle was detached from the nacre, and the entire soft body contracts toward the dorsal margin (Fig. 2A). Shell spines often developed in the infected group (Fig. 2C), and thus, the shell formation is probably normal until just before the disease onset. While the pearl oyster is sometimes atrophied due to low water temperatures in winter, no disease accompanied by atrophy during the period of high water temperatures has been reported until the occurrence of summer atrophy. Pearl oysters that have recovered from summer atrophy showed symptoms of so-called “shell disease” (Huang, Xie & Zhang, 2019), which is characterized by brownish deposit on the nacre and loss of nacre luster often in the areas close to the edge of the nacre (Figs. 2A–2C). It is speculated that the deposition of brownish substance occurs due to the formation of a prismatic layer on the surface of the nacre during the process of the exfoliated mantle returns to its original position (Sano, Kuriyama & Komaru, 2021). The prismatic layer is usually formed on the outer layer of the nacre. The loss of luster is probably the result of denaturation of the nacre by being exposed to seawater when the mantle is detached from the nacre surface. The abnormalities in the nacre were observed in 60–97% of the oysters that survived summer atrophy, according to a survey by the Mie Prefecture Fisheries Research Institute. Black-spot shell disease also causes aberrant pigmentation in the nacre, but the deposits are darker than those occurring with summer atrophy. Furthermore, deposition of black-spot shell disease typically occurs towards the dorsal edge (Fig. 2E), and death does not occur. Akoya oyster disease causes atrophy of the soft body, whereas the nacre remains normal (Fig. 2F), and mortality is high among adults (Iwanaga, Hirai & Hosokawa, 2008). Although mortality from marine birnavirus infection have been reported, the pathogenicity of the virus is relatively low (Suzuki, Utsunomiya & Kusuda, 1998).

Figure 2 Appearances of diseased and healthy pearl oysters.

(A) A 1-year-old pearl oyster with summer atrophy. The soft body is atrophied and is contracted towards the dorsal margin. The arrowhead indicates brown deposit on the nacre along the mantle edge. (B) Nacre of 2-year-old pearl oysters affected by summer atrophy. Arrowheads indicate brown deposit on the nacre. (C) Zero-year-old pearl oysters affected by summer atrophy. Deposition of brownish material is seen on the nacre surface (white arrowheads), whereas shell spines are extending (yellow arrowheads). (D) A normal pearl oyster. The mantle covers the entire nacre surface. (E) A pearl oyster with black-spot shell disease. Dark, organic matter is deposited mainly from the dorsal edge to the central part on the nacre. This individual was sampled from the group affected by summer atrophy. Loss of luster (the white arrowhead) and slight pigmentation (the yellow arrowhead) seem to be the symptoms of summer atrophy. (F) A Pearl oyster with Akoya oyster disease. The soft body is atrophied and has turned reddish-brown in color. No abnormality is observed on the nacre.

Environmental factors have been suspected of the cause of the disease, such as high water temperatures, decrease in food plankton, or chemical pollution. However, it is unlikely that these environmental changes caused summer atrophy, considering that the disease with the same symptoms occurred almost simultaneously in most of the pearl culture regions throughout Japan, where environmental conditions vary substantially. In addition, no abnormalities have been reported in organisms other than pearl oysters, such as other shellfishes and fish, in the areas where summer atrophy has been reported. On the other hand, the disease could be explained by an infectious agent, which might have spread in the first year from the original outbreak through the active transportation of juvenile oysters among culture areas, causing outbreaks almost simultaneously at a variety of locations across the country in the second year. In this study, we conducted infection tests to verify whether summer atrophy is a transmissible disease.

Materials & Methods

Animals

Details and localities of the pearl oysters, Pinctada fucata (Gould), used in each infection test are shown in Table 1 and Fig. 1. One-year-old, healthy pearl oysters were obtained from Ishikawa or Kanagawa Prefecture, where pearls are not cultivated (Fig. 1, Table 1). These oysters were used as recipients and negative control donors in the infection tests. The pearl oysters from Ishikawa Prefecture that were used as recipients in Test 2 showed slight pigmentation on the nacre in 2 of the 10 individuals observed, although the nature of pigmentation was much darker than those typically found in the oysters affected by summer atrophy and rather similar to that of black-spot shell disease. For the donors, shells exhibiting atrophy in the soft body were obtained from a pearl oyster farmer in Mie Prefecture (Table 1). Prior to the start of each experiment, 5 or 10 individuals were randomly sampled from each batch to examine atrophy of the soft body and nacre abnormality. Oysters of each experimental group were reared in the same 65-L tank and different groups were separately reared in different 65-L tanks. The tanks were filled with 56 L running seawater (approximately 250 mL/min) filtered with a one-μm pore size filter and maintained at 23–25 °C under the natural photoperiod in the laboratory. For food, 500 mL of cultured diatoms (Chaetoceros neogracile, approximately 5 × 105 cells/mL), was added to each aquarium five times per week.

Table 1 Summary of pearl oysters used.

Infection experiments	Collection sites	Age (years old)	Range in shell width (mm, mean ± SD)	Prevalence of atrophy#	Nacre abnormalities*	Mean shell score	Source of inoculum	Note	
Healthy pearl oysters (Recipients)							
Test 1	Ishikawa	1	45–57
(49.3 ± 3.7)	0/10	0/10	0 ± 0			
Test 2	Ishikawa	1	51–63
(59.6 ± 3.1)	0/10	2/10	0.4 ± 0.3			
Test 3	Kanagawa	1	45–56
(49.8 ± 3.6)	0/5	0/5	0 ± 0			
Test 4	Kanagawa	1	55–68
(61.8 ± 2.3)	0/10	0/10	0 ± 0			
Test 5-1	Kanagawa	1	ditto	0/10	0/10	0 ± 0			
Test 5-2	Kanagawa	1	51–64
(55.0 ± 5.6)	0/5	0/5	0 ± 0			
Test 6-1	Kanagawa	2	51–65
(57.6 ± 1.4)	0/5	0/5	0 ± 0			
Test 6-2	Kanagawa	3	50–75
(15.8 ± 0.8)	0/5	1/20	0.1 ± 0.1			
Source of infection (Donors)							
Test 1	Mie	3	73–60
(65.0 ± 1.5)	3/10	7/10	2.3 ± 0.6	Hemolymph	There are traces of previous year.	
Test 2	Mie	0	NM	5/10	0/10	0 ± 0	Supernatant of macerated tissue	Cumulative mortality rate at the time of sampling was ca. 20%. No deaths up to 6 days before sampling.	
Test 3	Mie	0	27–32
(29.2 ± 1.2)	2/10	4/10	1.3 ± 1.1	Supernatant of macerated tissue	Nine days before the experiment, 9 out of 10 individuals were atrophied. Mortality rate unknown.	
Test 4	Mie	0	26–45
(33.5 ± 2.0)	2/10	NM	NM	Supernatant of macerated tissue	The same lot as in Test 3 were used.	
Test 5-1	Mie	0	28–41
(32.0 ± 3.3)	2/10	NM	NM	Supernatant of macerated tissue	The same lot as in Test 3 were used.	
Test 5-2	Experimentally infected	1	50–67
(58.6 ± 3.1)	4/5	0/5	0 ± 0	Hemolymph	Obtained from Test 4	
Test 6-1	Experimentally infected	1	52–61
(56.0 ± 2.6)	1/3	0/3	0 ± 0	Hemolymph	Obtained from Test 4	
Test 6-2	Mie	0	12–20
(15.8 ± 0.8)	4/10	7/10	1.7 ± 0.5	Supernatant of macerated tissue	Cumulative mortality rate at the time of sampling was ca. 33.3%. No deaths up to 2 days before sampling.	
Notes:

* The number of oysters in which nacre abnormality was observed/the number of oysters sacrificed from each group immediately before the start of the experimets. No atrophy in the soft tissues was observed in the recipient groups.

# The number of oysters in which the soft body was atrophied/the number of oysters sacrificed from each group immediately before the start of the experiments. Nacre abnormalities remain after the oyster recovered from the disease, and hence, it cannot be used to judge if the oyster is diseased at the time of observation.

NM, not measured.

Scoring degree of disease

Although the infection tests were performed using 1-year-old adult oysters in this study, the mortality by summer atrophy of shells over 1-year-old is low. Consequently, severity of the disease in the infection tests could not be assessed by mortality. The atrophy of the soft body is characteristic of the disease. However, the atrophy occurs only transiently at different times in different individual oysters, and shells must be opened when the atrophy is most severe, to determine the severity of the disease. This is intrinsically difficult. On the other hand, observations of the valves of spontaneously diseased oysters suggest that the deposition of brown pigment and loss of nacre luster are also characteristic of this disease. These nacre abnormalities remain long after the oysters recover from the disease, so that the shells can be opened at any arbitrary timing to determine the severity of the disease. Therefore, in this study, the degree of disease severity was determined using abnormalities in the nacre.

The extent of abnormalities in the nacre was quantified with a subjective index as shown in Fig. 3. For the infection tests, the scores for the two valves of an individual were totaled for each pearl oyster, and the sum of the scores of oysters in one group was divided by the number of tested oysters in the group to obtain the mean shell score of the group. For the investigation of the nacre of farmed oysters that was affected by spontaneously occurring summer atrophy, the sum of the scores of valves for each group of oysters was divided by the number of observed valves to obtain the mean shell score of the group. This is because the two valves of a stored individual oyster had usually fallen apart and reconstruction of an individual by two valves was difficult.

Figure 3 Scoring the severity of disease.

The severity of the disease was quantified according to the ratio of abnormal areas (deposition of brown pigment and loss of luster) to the total nacreous surface: score 0, normal; score 1, the abnormal area is less than 30%; score 2, the abnormal area is larger than 30% and less than 70%; and score 3, the abnormal area is larger than 70%. In these photographs, the shell with score 1 has brown deposit (the white arrowheads) on the upper right, and the shell with score 2 has brown deposit on the upper left and bottom. In the shell with score 3, deposition was observed on the upper and lower areas where the mantle was located, and the surrounding large area of nacre has lost its luster (the yellow arrowheads).

Investigation of spontaneously occurring disease in farmed oysters

Atrophy of the soft body and abnormalities in the nacreous layer were investigated in cultured and wild pearl oysters. Cultured oysters were obtained from the areas where summer atrophy is occurring, and wild, disease-free oysters were from the areas where the disease has not been reported (Table 2). We also observed the nacreous layer of stored valves of cultured pearl oysters. These oysters had been collected in Mie prefecture before June 2019, prior to the first outbreak of summer atrophy.

Table 2 Prevalence of atrophy and nacre abnormalities of the pearl oysters collected in different regions and ages.

Collection dates	Collection sites	Age (years old)	Range in shell width (mean ± SD)	Prevalence of atrophy*	Nacre abnormalities/valve#	Mean shell score/valve	Note	
Summer atrophy-free area	
2019.8.23	Ishikawa	1	49–67
(58.5 ± 1.0)	0/40	0/80 (0%)	0 ± 0		
2020.6.4	Ishikawa	1	42–66
(56.0 ± 1.8)	0/40	2/40
(5%)	0.05 ± 0.04		
2019.10.22	Wakayama	0-2	29–55
(44.6 ± 2.5)	0/16	0/32	0 ± 0		
2019.11.1	Wakayama	0-2	29–55
(44.6 ± 2.5)	0/20	0/40	0 ± 0		
2020.9.10	Kanagawa	1	45–60
(58.8 ± 5.8)	0/30	1/60† (1.7%)	0.03 ± 0.02		
2020.10.21	Kanagawa	1	51–73
(61.9 ± 3.8)	0/30	0/60	0 ± 0		
Summer atrophy-epidemic area	
Before summer atrophy epidemic	
2014.4.15 !	Mie	2	53-71
(59.8 ± 1.2)	NM	0/19	0 ± 0		
2014.7.19 !	Mie	2	53-64
(53.3±2.3)	NM	0/42	0 ± 0		
2017.4.21 !	Mie	1	39-54
(46.2±0.6)	NM	1/82† (2.4%)	0.05 ± 0.05		
2017.5.1!	Mie	1	60-74
(64.2 ±1.1)	NM	1/17† (0.6%)	0.1 ± 0.1		
2019.4.25!	Mie	1						
After summer atrophy epidemic	
2019.8.23	Mie	1	41–55
(48.5 ± 1.7)	5/10	16/20 (80%)	1.9 ± 0.4	The cumulative mortality rate of 0-year-old oysters at the end of the mass mortality was more than 50%.	
2019.9.19	Mie	0	27- 42
(34.8 ± 0.7)	NM	17/30 (56.7%)	1.2 ± 0.3	Randomly sampled, independent of disease onset.	
2019.10.24	Mie	0 and 1	14–58
(29.9 ± 3.0)	NM	31/38 (81.6%)	1.6 ± 0.2	The cumulative mortality rate of 0-year-old oysters at the end of the mass mortality was ca.70%.	
2019.11.20	Mie	1	50–57
(53.6 ± 0.5)	NM	11/20 (55%)	1.3 ± 0.3	Randomly sampled, independent of disease onset.	
2020.3.11	Mie	1	50–68
(56.1 ± 1.4)	0/10	13/20 (65%)	1.1 ± 0.3	Randomly sampled, independent of disease onset.	
2020.6.24	Mie	1	53–69
(59.0 ± 0.6)	7/44	24/44 (54.5%)	0.9 ± 0.2	Obtained a few days after the pearl farmer found the atrophy. Cumulative mortality rate at the time of sampling was ca. 20%.	
2021.6.23	Mie	1	25–66
(47.8 ± 1.8)	NM	22/36 (61.1%)	1.1 ± 0.2	Obtained a few days after the pearl farmer found the atrophy.	
2019.8.29	Ehime	1	24–58
(39.5 ± 2.3)	NM	33/48 (68.8%)	1.3 ±0.2	The cumulative mortality rate of 0-year-old oysters reared at the farm at the time of sampling was ca. 33.3%.	
2019.10.17!	Ehime	0	26–45
(39.4 ± 0.4)	NM	39/68 (45.6%)	1.2 ± 0.2	Randomly sampled, independent of disease onset.	
2019.10.17!	Ehime	1	50–70
(59.9 ± 0.6)	5/30	36/60 (60%)	1.4 ± 0.2	Randomly sampled, independent of disease onset.	
2020.9.17!	Ehime	1-2	51–75
(61.3 ± 2.8)	NM	7/10 (70%)	1.4 ± 0.4	Randomly sampled, independent of disease onset.	
Notes:

* The number of oysters in which the soft body was atrophied/the number of oysters sacrificed from each group immediately before the start of the experiments. Nacre abnormalities remain after the oyster recovered from the disease, and hence, it cannot be used to judge if the oyster is diseased at the time of observation.

# The number of valves in which nacre abnormality was observed/the number of tested valves from each group.

! Valves stored in a dry state at room temperature were observed on 2021.7.6.

† Nature of pigmentation was much darker than those typically found in the oysters affected by summer atrophy and rather similar to that of black-spot shell disease.

Nacre abnormalities and mean shell score were calculated for the incidence and score per shell, not per individual.

NM, not measured

Test 1: cohabitation and injection challenge using adult pearl oysters as the infection source

In the cohabitation test, five diseased 3-year-old oysters and 14 healthy oysters were reared in different net cages set in a 65-L tank under the conditions described above so that the diseased and healthy oysters are not in contact with each other. For the negative control, 19 healthy oysters were reared similarly in another 60-L tank. In the injection test, the hemolymph of 15 diseased oysters was collected from the adductor muscles. The hemolymph was pooled and centrifuged at 1,200×g for 10 min to obtain the supernatant, which was used as the inoculum. This inoculum was injected into the adductor muscles of 14 healthy pearl oysters (100 μl/individual) with 23 × 1/4 gauge needles. For the negative control, 14 healthy oysters were inoculated with hemolymph into the adductor muscles using supernatant prepared from three healthy oysters by the above method. Experiments were performed at two different water temperatures, 25 °C and 22–23 °C. That is, a total of eight test groups were set (Table 2). The mortality was monitored during the experiment. Since the abnormality on the nacre seemed to take several days to appear and mortality sometimes occurred early in the experiments, the shell scores were determined only on surviving specimens 27 days after the start of each experiment.

Test 2: injection challenge using juvenile pearl oysters as the infection source

Soft bodies were collected from 12 0-year-old juveniles from an affected group. They were pooled (0.41 g in total wet weight), chopped with scissors in two mL of autoclaved seawater, and centrifuged at 1,200×g for 10 min to obtain the supernatant as the inoculum. The inoculum for the negative control was prepared in the same way from healthy pearl oysters (0.43 g in total wet weight). Each inoculum was injected into the adductor muscles of 10 healthy oysters (100 μl/individual). The mortality was monitored, and the shell scores were determined on surviving individuals 18 days after the injections.

Test 3: filterability

Soft bodies were collected from 20 diseased 0-year-old juveniles, pooled (4.9 g in total wet weight), and cut into small pieces with scissors. The volume of the minced tissues was adjusted to 50 mL with autoclaved seawater and centrifuged at 1,200×g for 10 min to obtain the supernatant. The supernatant was then sequentially filtered through 0.8, 0.45, 0.22 and 0.1 μm filters (Millipore, Millipore Co., Tokyo, Japan). Part of each filtrate was saved and inoculated to the adductor muscles of seven healthy oysters (100 μl/individual). Thus, the total of 28 shells were used. For the positive control, diseased oysters (N = 5) and healthy oysters (N = 7) were reared in the same aquarium as in Test 1, since it had been known that the disease could be transmitted through cohabitation by the time when this test was conducted. Two negative control groups were also prepared. For one group, only healthy pearl oysters (N = 7) were reared in an aquarium. For the other, supernatant of the soft body (4.86 g) from a single healthy animal was prepared as described above and injected into the adductor muscles of seven healthy oysters (100 μl/individual) without filtration. Experiments were performed in duplicate in all test groups including positive and negative controls. The mortality was monitored, and the shell scores were determined on surviving individuals 38 days after the start of the experiment.

Test 4: serial passages

To further clarify whether the disease was transmissible, we performed an experiment to know whether the disease could be passed on repeatedly. The serial passages were conducted with three different intervals: 1 h, 5 and 10 days. The outline of the test is shown in Fig. 4. Soft bodies collected from 10 0-year-old oysters of an affected group were pooled (wet weight 4.0 g) and cut into small pieces with scissors. The volume of the minced tissues was adjusted to 6 mL with autoclaved seawater and centrifuged at 1,200×g for 10 min. The supernatant was filtered through a 0.1 um filter (Millipore). The filtrate was injected into the adductor muscles of 25 healthy oysters (100 μl/individual); this infection test was defined as the first passage. From the 25 oysters, five individuals were randomly selected at 1 h, 5 and 10 days after the inoculation, and the hemolymph was collected and pooled to prepare the inoculum for the second passage. The pooled hemolymph was centrifuged at 1,200×g for 10 min to obtain the supernatant as the second inoculum. The inoculum was injected into the adductor muscles of 12 healthy pearl oysters (100 μl/individual) in each test series (total N = 36). As for the third passage, five individuals were selected from each group at 1 h, 5 and 10 days after the second inoculation, and the hemolymph was collected and used as the inoculum. This procedure was repeated up to the 3rd passage for the 1-h interval passages, to the 4th passage for the 5-day interval passages, and to the 8th passage for the 10-day interval passages. The remaining oysters in each passage (10 individuals for the first passage and seven for the other passages) were kept for 3 weeks to observe shell abnormality. For negative controls, seven healthy oysters were similarly inoculated with supernatant of hemolymph that was prepared from five healthy animals. Serial passages were not conducted for the negative controls.

Figure 4 Outline of the transmission experiments with serial passages.

The serial passages were conducted with 3 different intervals: 1 hour, 5 days, and 10 days. Hemolymph was collected from 5 specimens in each cohort, pooled, and injected into the next group. The remaining 7 individuals in each cohort were sampled 3 weeks after inoculation to observe abnormalities of nacre surface.

Test 5: chloroform sensitivity

This test was performed twice (Table 1). As the inoculum in the first test (Test 5-1), the supernatant of the hemolymph used for the infection of the first passage in Test 4 was used. In the second test (Test 5-2), the supernatant of the hemolymph used for the third passage in the 10-day interval passages in Test 4 was used. In both tests, the inoculum was divided into two equal aliquots, an equivalent volume of chloroform was added to one aliquot, and the mixture was shaken at 25 °C at 300 rpm for 10 min and then centrifuged at 1,200×g for 10 minutes to recover the aqueous layer as the chloroform-treated inoculum. The other was similarly shaken and centrifuged without adding chloroform and used as the non-treated inoculum. For the negative control, supernatant of the hemolymph (centrifuged at 1,200×g for 10 min) of a single healthy oyster was used as the inoculum. Each inoculum was injected into the adductor muscles of five and seven healthy pearl oysters in the Tests 5-1 and 5-2, respectively (100 μl/individual). The mortality was monitored during the experiments. The shell scores were determined on the surviving individuals 15 and 20 days after the injections in the two tests, respectively.

Test 6: thermal stability

This test was performed twice (Table 1). In the first test (Test 6-1), the supernatant of the hemolymph used for the 7th passage in the 10-day interval passages in Test 4 was used as the inoculum. In the second test (Test 6-2), the inoculum was prepared from 0-year-old juveniles of an affected group (Table 1); soft bodies were collected from 10 juveniles, pooled (0.8 g in total wet weight), chopped with scissors in 5 mL of autoclaved seawater, and centrifuged at 1,200×g for 10 min to obtain the supernatant as the inoculum. For each experiment, the inoculum was dispensed into 1.5 mL tubes (1.0 mL/tube). The tubes were incubated for 30 min in heat blocks maintained statically at four different temperatures: 40, 50, 60 °C, or on ice. Subsequently, all tubes were rapidly cooled in an ice bath. For the negative controls, supernatant was prepared from hemolymph (Test 6-1) or macerated tissue (Test 6-2) of healthy animals and used without heat treatment. Each inoculum was injected into the adductor muscles of five healthy pearl oysters (100 μl/individual). The mortality was monitored during the experiments. The shell scores were determined on the surviving individuals 21 and 15 days after the injections in Tests 6-1 and 6-2, respectively.

Histopathology

Sampling for histopathology was conducted on days 5 and 10 after the inoculation of the first passage in the Test 4. For the 5-day samples, tissues were taken from the oysters used as donors for the 2nd passage for 5-day interval passages. For the 10-day samples, tissues were taken from the animals used as donors for the 2nd passage for 10-day interval passages. At each sampling, the mantle, gill, kidney, heart, adductor muscle, foot, and midgut (including the stomach and gonads) were dissected from five specimens after collecting the hemolymph. The tissues were fixed in Davidson’s solution (Bell & Lightner, 1988), dehydrated through an ethanol series and embedded in paraffin. Two sets of three μm sections were prepared from each tissue, stained with hematoxylin and eosin or May-Grunwald/Giemsa, and observed under a microscope.

Statistics

The mean shell scores were expressed as means ± standard error. The mortality and prevalence of nacre abnormalities in survivors were analyzed statistically with Fisher’s exact test, and the mean shell scores were tested by Mann–Whitney U-test or the Steel-Dwass test. A value of p < 0.05 was considered to be statistically significant.

Results

Investigation of spontaneously occurring disease in farmed oysters

In the oysters obtained from pearl farmers in Mie and Ehime prefectures where summer atrophy occurred, nacre abnormalities were observed in 45.6∼81.6% of the oysters (Table 2). Cultured pearl oysters from Mie prefecture before the disease occurred, and wild pearl oysters from areas where pearl farming has never performed and the disease has not been recorded, did not show the nacre abnormalities that are seen in the disease.

Test 1: cohabitation and injection challenge using adult pearl oysters as the infection source

There were no significant differences in mortality between test groups (Table 3). Appearances of nacre abnormalities of the experimental groups were significantly higher than those of the corresponding negative controls, except for the cohabitation experiment at the low temperature (Table 3). Shell scores of the injection-challenged groups were significantly higher than those of the negative controls.

Table 3 Test 1: Cohabitation or injection with infected adult pearl oysters as the source of infection at two water temperatures.

	Infection experiments	Negative controls	
	Cohabitation	Injection	Cohabitation	Injection	
Water temperature　22–23°C	
Mortality
(dead/examined)	1/14	2/14	2/14	1/14	
Nacre abnormalities!	2/13	9/12*	1/12	1/13	
Mean shell scores of survivors	0.5 ± 0.3	2.8 ± 0.4#	0.3 ± 0.1	0.07 ± 0.07	
Water temperature 25°C	
Mortality
(dead/examined)	0/14	1/14	1/14	0/14	
Nacre abnormalities!	5/14*	9/13*	1/13	0/14	
Mean shell scores of survivors	0.9 ± 0.4	2.3 ± 0.4#	0.07 ± 0.07	0 ± 0	
Notes:

* Significantly different from the negative control group (Fisher’s exact test p < 0.05).

# Significantly different from the negative control group (Mann–Whitney U-test p < 0.05).

! The number of oysters in which nacre abnormality was observed/the number of surviving oysters.

Test 2: injection challenge using juvenile pearl oysters as the infection source

No mortality was observed in the negative control group (Table 4). In the group injected with inoculum prepared from diseased pearl oysters, death of the experimental animals started on the 11th day after inoculation, and 7 out of 10 individuals died by the end of the experiment. Six of the seven dead specimens exhibited shell abnormalities, and the mean shell score obtained for the seven dead individuals was 1.7 ± 0.4. The abnormalities on the nacre were also confirmed in all three surviving individuals, and the mean shell score for the three was 3.3 ± 0.7. No abnormalities were observed in the shells of the surviving individuals in the negative control group.

Table 4 Test 2: Injection with juvenile pearl oysters as the source of infection.

	Infection experiments	Negative controls	
Mortality
(dead/examined)	7/10*	0/10	
Nacre abnormalities of dead!	6/7	–	
Nacre abnormalities of survivor!	3/3	0/10	
Mean shell scores of dead oysters	1.7 ± 0.4	–	
Mean shell scores of survivors	3.3 ± 0.7	0 ± 0	
Notes:

* Significantly different from the negative control group (Fisher’s exact test p < 0.05).

! The number of oysters in which nacre abnormality was observed/the number of observed oysters.

Test 3: filterability

The results are summarized in Table 5. Mortality was only found in the experimental and positive control groups. Nacre abnormalities were also mostly observed in the experimental and positive control animals, although slight abnormalities were observed in some control oysters. Thus, the disease was clearly reproduced only in the experimental and positive control groups including the experimental oysters injected with the hemolymph filtered through 0.1 μm filter.

Table 5 Test 3: Filterability of causative agent of summer atrophy of pearl oyster.

	Pore size of membrane filters (μm) used for filtration of the hemolymph of affected oysters	Negative controls	Positive control	
	0.8	0.45	0.22	0.1	Cohabitation with healthy oysters	Injection with healthy oyster hemolymph	Cohabitation with affected oysters	
Experimental series-1	
Total number of deaths	1	0	1	1	0	0	1	
Nacre abnormalities!	6/6*	7/7*	6/6*	5/6*	0/7	0/7	5/6*	
Mean shell scores of survivors	3.3 ± 0.7#	3.9 ± 0.8#	4.2 ± 0.5#	3.0 ± 0.8#	0 ± 0	0 ± 0	2.8 ± 0.9#	
Experimental series-2	
Total number of deaths	2	1	3	1	0	0	2	
Nacre abnormalities!	5/5*	6/6*	3/4*	6/6*	2/7	0/7	5/5*	
Mean shell scores of survivors	3.4 ± 0.7#	2.5 ± 0.5#	3.3 ± 1.1#	4.3 ± 0.2#	0.3 ± 0.2	0 ± 0	3.2 ± 0.5#	
Notes:

* Significantly different from healthy oyster hemolymph—injected group (Fisher’s exact test, p < 0.05).

# Significantly different from healthy oyster hemolymph—injected group in the same experimental series (Steel-Dwass test, p < 0.05).

! The number of oysters in which nacre abnormality was observed/the number of surviving oysters.

Test 4: serial passages

Neither mortality nor nacre abnormalities were observed in the negative control group (Table 6). In the 1-h interval series, shell abnormalities were observed by the second passage, but no abnormalities were observed in the 3rd passage. On the other hand, in the 5 and 10-day interval passages, the nacre abnormalities were observed up to the end of the experiments.

Table 6 Test 4: Mortality and disease appearances in serial passages.

No. of passages	Dead/examined	Nacre abnormalities!	Mean shell scores of survivors	Dead/examined	Nacre abnormalities!	Mean shell scores of survivors	
	Negative control	1-hour passage series	
1	0/7	0/7	0 ± 0	0/10	10/10*	3.9 ± 0.6#	
2				0/7	6/7*	3.0 ± 0.6	
3				0/7	0/7	0 ± 0	
	5-days passage series	10-days passage series	
1	0/10	10/10*	3.9 ± 0.6#	0/10	10/10*	3.9 ± 0.6#	
2	0/7	7/7*	4.4 ± 0.5#	0/7	7/7*	5.0 ± 0.5#	
3	0/7	6/7*	3.1 ± 0.8#	3/7	3/4*	1.8 ± 0.6	
4	0/7	6/7*	3.3 ± 0.7#	0/7	6/7*	1.9 ± 0.6	
5				0/7	6/7*	3.5 ± 0.8#	
6				NM	NM	NM	
7				NM	NM	NM	
8				1/7	7/7*	4.0 ± 0.9#	
Notes:

* Significantly different from a negative control (Fisher’s exact test, p < 0.05).

# Significantly different from a negative control (Steel-Dwass test, p < 0.05).

! The number of oysters in which nacre abnormality was observed/the number of surviving oysters.

Blanks indicate that the experiments were not conducted. NM: not measured Passage 1 is common to all of the passage test series performed (i.e., every hour, 5 days, and 10 days).

Test 5: chloroform sensitivity

In both experiments, the nacre abnormalities appeared similarly in the two experimental groups irrespective of the treatment of chloroform, whereas no abnormalities were found in the nacre of the control oysters (Table 7).

Table 7 Test 5: Sensitivity to chloroform of causative agent.

	Test 5-1	Test 5-2	
Inoculum	Nacre abnormalities!	Mean shell score of survivors	Nacre abnormalities!	Mean shell score of survivors	
Chloroform-treated	5/5	2.8 ± 0.9*	3/7	1.0 ± 0.6	
Non-treated	5/5	3.6 ± 0.8*	3/7	1.4 ± 0.8	
Hemolymph from healthy pearl oyster	0/5	0 ± 0	0/5	0 ± 0	
Notes:

* Significantly different from the group injected with healthy pearl oyster hemolymph (Steel—Dwass test, p < 0.05).

! The number of oysters in which nacre abnormality was observed/the number of surviving oysters.

Test 6: thermal stability

No abnormalities in the nacreous layer were observed in the groups injected with the inoculums incubated at 50 and 60 °C in Test 6-1, and in the group injected with the inoculum treated with heat at 60 °C in Test 6-2 (Table 8).

Table 8 Test 6: Thermal stability of causative agent of summer atrophy of pearl oyster pathogenicity.

	Temperature	Negative control	
	On-ice	40 °C	50 °C	60 °C	Injection with healthy oyster hemolymph	
Test-1						
Nacre abnormalities!	5/5*	4/5*	0/5	0/5	0/5	
Mean shell scores of survivors	2.0 ± 0.4#	2.0 ± 0.6	0 ± 0	0 ± 0	0 ± 0	
Test-2						
Nacre abnormalities!	4/5*	3/5*	4/5*	0/5	0/5	
Mean shell scores of survivors	2.8 ± 0.9	2.2 ± 0.8	2.4 ± 0.8	0 ± 0	0 ± 0	
Notes:

* Significantly different from healthy oyster hemolymph-injected group (Fisher’s exact test, p < 0.05).

# Significantly different from healthy oyster hemolymph-injected group (Steel-Dwass test, p < 0.05).

! The number of oysters in which nacre abnormality was observed/the number of surviving oysters.

Histopathology

In the oysters sampled on day 10, increased numbers of migrating cells were observed in the connective tissues in 4 out of 5 individuals, but no other significant histopathological changes were found in any of the observed specimens.

Discussion

The results of this study suggest that summer atrophy of Akoya pearl oyster is an infectious disease. The nacre abnormalities found in spontaneously affected oysters were reproduced in healthy pearl oysters that were reared with diseased oysters in 3 of the 4 experiments (Tests 1 and 2). Nacre abnormalities were also confirmed in the surviving oysters in all six infection tests with injections of the inoculum prepared from affected pearl oysters.

The mass mortality of juveniles and abnormalities in adult oysters in pearl culture farms are likely to be caused by the same causative agent, considering the results of the present study, in which the disease was reproduced either by affected adults or by affected 0-year-old juveniles as the donors. Nevertheless, infection tests on juveniles should be conducted in future studies to see if mass mortality could be reproduced, since only adult oysters were used as recipients in this study.

The results of this study also suggest that the etiology of this disease is proliferative. Both in the 5- and 10-day interval passages, the nacre abnormalities were confirmed up to the end of the experiments. Therefore, the etiological agent of this disease probably proliferates within 5–10 days after inoculation to the amount that is enough to cause the disease in the next passage. In the 1-h interval passages, however, none of the individuals in the third passage developed the abnormalities of nacre surface, although the nacre abnormalities were confirmed up to the second passage. The causative agent probably could not sufficiently propagate in 1 h, and the disease appeared in the second passage was most likely caused by the causative agent that had been carried over from the injection in the first passage.

The etiological agent in the host body appears to decrease quickly as the oyster recovers. In previous studies, we could not reproduce the disease either by cohabitation or injection when pearl oysters with normal soft bodies were used as donors, despite that we obtained those donors from the group that had experienced summer atrophy and confirmed typical nacre abnormalities in those animals (Tables S1 and S2). Therefore, to reproduce the disease, it seems important to use diseased oysters with atrophied soft bodies or oysters from populations that are currently experiencing mass mortality as the infection source.

Considering the filterability (Test 3), the sensitivity to chloroform (Test 5) and the thermal stability (Test 6) of the pathogen, the causative agent for summer atrophy seems to be a non-enveloped virus with a minor axis of less than 100 nm, which is inactivated 50 or 60 °C for 30 min. In addition, the optimal temperature for the propagation of the pathogen might be around 25 °C or more, since the disease was not clearly reproduced by cohabitation at 22–23 °C in this study. This is supported by the fact that the spontaneous disease outbreaks occur during summer as stated before. However, this requires further study.

We could not find any noticeable histopathological changes in the affected pearl oysters in the present study. In fact, we have not observed any substantial histopathological changes that are common to spontaneously diseased oysters presumably affected with summer atrophy so far. This seems peculiar for a disease with mass mortality caused by an infectious agent. It is possible that we have overlooked some important histological changes, although it would be clarified if the pathogen is definitively identified.

Several cases similar to summer atrophy have been reported in the genus Pinctada, but none are identical to the disease. Birnavirus (Suzuki, Kamakura & Kusuda, 1998) and “Akoya virus” (Miyazaki et al., 1999) were reported to infect P. fucata in Japan, although the symptoms of summer atrophy were not observed with those virus infections. Moreover, neither biochemical nor morphological characteristics have been reported for Akoya virus, and its existence itself is uncertain. Akoya virus was reported to be the causative agent of Akoya oyster disease (AOD), although its relationship with AOD has been refuted (Matsuyama et al., 2017). Virus infections have also been reported in P. maxima (Pass, Perkins & Dybdahl, 1988; Norton, Shepherd & Prior, 1993). However, the characteristic viral inclusion bodies observed in these studies were not found in this study, and hence, these viruses seem to be unrelated to summer atrophy of P. fucada.

Mass mortalities of unknown etiology has been reported in the members of the genus Pinctada (Wolf & Sprague, 1978, Nasr, 1982, Comps et al., 2001, Jones et al., 2010). One of those, “syndrome 85” is similar to summer atrophy in P. fucada, in that juvenile experience mass mortality and show symptoms that include deposition of brown organic matter on the inner surface of the two valves in P. margaritifera (Comps et al., 2001). Unlike summer atrophy, however, high mortality occurs irrespective of the age of the shells in syndrome 85, and the disease is also characterized by granulomatous tissue formation (Comps et al., 2001). Another abnormal condition with unknown etiology is oyster oedema disease (OOD). The disease caused mass mortality in P. maxima in Western Australia in 2006 (Jones et al., 2010). Similar to summer atrophy, the mantle of OOD-affected oysters is contracted, and histological responses are not observed (Jones et al., 2010). However, edema does not occur in summer atrophy. Thus, summer atrophy of P. fucada seems to be a new infectious disease in the genus Pinctada, although it is possible that the differences in the symptoms in those diseases with unknown etiology in Pinctada might be due to the difference in the host species.

Abnormalities in shells of bivalves, which are often collectively called “shell disease”, can be caused by a variety of organisms, such as parasites (Martinelli et al., 2020; Farley, 1968), fungi (Friedman, Grindley & Keogh, 1997; Raghukumar & Lande, 1988; Renault et al., 2002), bacteria (Elston & Leibovitz, 1980; Paillard & Maes, 1995; Elston, Frelier & Cheney, 1999; Boettcher et al., 2005; Huang, Xie & Zhang, 2019), and viruses (Momoyama, Nakatsugawa & Yurano, 1999; Matsuyama et al., 2020). Shell disease in P. fucata has also been experimentally reproduced by inoculating viable Escherichia coli or the yeast Saccharomyces cerevisiae into the extrapallial space of the valves (Huang, Xie & Zhang, 2019). Since the responsible tissue for shell formation is the mantle, these pathogens probably affect the activity of mantle, either directly or indirectly, and lead to shell disease. Sano, Kuriyama & Komaru (2021) compared the expression levels of five genes in the mantle of healthy P. fucata and the animals affected with summer atrophy, and found decreased expression of the OT47 gene (tyrosinase gene involved in the formation of periostracum (Zhang et al., 2006)) and increased expression of the msi31 gene (shell matrix protein gene involved in the formation of the prismatic layer (Sudo et al., 1997)) in the diseased oysters. The expression levels of these genes may be altered by the causative agent of the disease in the mantle. However, no obvious histopathological change has been observed in the tissues of the diseased pearl oysters, including the mantle, so far. It is therefore unknown whether the etiological agent of the disease exists in the mantle. To resolve these issues and to identify the etiological agent, we are currently conducting experiments utilizing next-generation sequencing.

Conclusions

This study was conducted to clarify the involvement of infectious agents in the mass mortality in juveniles and atrophy and mortality in adult pearl oysters (P. fucata) in the major pearl farming areas in Japan in 2019 and 2020. The condition, or the disease, is referred to as “summer atrophy” here. The oysters recovered from the disease exhibited deposition of brown materials and loss of luster on the inner nacreous surface of valves, which is usually smooth and shiny. These shell abnormalities were reproduced in healthy pearl oysters either by injecting hemolymph from affected oysters or by cohabitation with diseased specimens. The disease was successfully reproduced through continual passages using hemolymph as the inoculum up to the 8th passage. The cause of this disease is hence considered to be a proliferative pathogen. In addition, pathogenicity was not affected by filtering the inoculum through a 0.1 μm filter or by chloroform treatment. Therefore, the etiological agent is likely to be a non-enveloped virus with a diameter of less than 100 nm. Definitive identification of this pathogen is the ongoing subject of our study.

Supplemental Information

Supplemental Information 1 Infection test by cohabitation with affected pearl oysters.

Click here for additional data file.

Supplemental Information 2 Infection test by hemolymph injection.

Click here for additional data file.

Supplemental Information 3 Supplemental Tables with raw data.

Shell scores of each survived pearl oysters.

Click here for additional data file.

The authors would like to thank Mr. I. Kuriyama, Mie Prefecture Fisheries Research Institute, Dr. H. Kawakami and Dr. T. Itano, Ehime Prefectural Fish Disease Control Center and Dr. S. Shirakashi, Kindai University for their assistance for sampling pearl oyster.

Additional Information and Declarations

Competing Interests

Author Contributions

Data Availability

The authors declare that they have no competing interests.

Tomomasa Matsuyama conceived and designed the experiments, performed the experiments, analyzed the data, prepared figures and/or tables, authored or reviewed drafts of the paper, and approved the final draft.

Satoshi Miwa performed the experiments, analyzed the data, prepared figures and/or tables, authored or reviewed drafts of the paper, and approved the final draft.

Tohru Mekata performed the experiments, authored or reviewed drafts of the paper, and approved the final draft.

Yuta Matsuura performed the experiments, authored or reviewed drafts of the paper, and approved the final draft.

Tomokazu Takano performed the experiments, authored or reviewed drafts of the paper, and approved the final draft.

Chihaya Nakayasu conceived and designed the experiments, analyzed the data, authored or reviewed drafts of the paper, funding acquisition, and approved the final draft.

The following information was supplied regarding data availability:

The raw measurements are available in the Supplementary Files.

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
