# Peer review of "Mass mortality of pearl oyster (Pinctada fucata (Gould)) in Japan in 2019 and 2020 is caused by an unidentified infectious agent"

_PeerJ, doi:10.7717/peerj.12180_

## Round 0.1 · original submission · Major Revisions

The reviewers have highlighted the strengths of your submitted manuscript and have proposed a number of changes to improve the quality.

Please consider the proposed changes as mandatory for the final acceptance of your manuscript. This is particularly relevant for the second major comment of reviewer 1 and the statistical aspect raised by reviewer 2.

I look forward to your revised manuscript.

Reviewer 1 ·

Basic reporting

no comment

Experimental design

no comment

Validity of the findings

no comment

Additional comments

This paper is devoted to identification of the causative agent of mass mortality in 0-year-old pearl oysters and anomalies in adults which occurred in Japan in the summer of 2019 and 2020. The incidence was named as “summer atrophy” to discriminate against the diseases reported previously such as Akoya oyster disease and marine birnavirus infection. Based on the filterability and sensitivity to chloroform of the pathogen, the causative agent was suggested to be a non-enveloped virus. Although this paper has potential contribution to the progress of our knowledge in marine science and aquaculture, some issues remain to be considered carefully as follows.

Major comments:
1) It seems reasonable to conclude that the causative agent is a non-enveloped virus considering the results obtained in this paper. However, it may be more confirmative if the authors set up experiments with pathogenic samples after heat treatment to examine whether or not pathogenicity remains.
2) It is curious about no apparent histopathological effects (lines 221 – 223). Figures 2 and 3 show severe changes in the structure of mantle accompanied by the formation of black spots and the loss of luster on the nacre. Is it relevant to have observed migrating cells in the connective tissues (probably in mantle)?

Others
Line 44: peaking -> attaining the peak.
Line 49: improve the sentence.
Line 64: improve the name of Fisheries Research Institute.
Line 73: and/or.
Line 91: running speed? Day and night period.
Line 106: explain 3+.
Line 182: examined -> observed.
Line 195: Why are the results not shown in Table?
Line 208: um -> (micro)m.
Line 267 – 269: reference is needed.
References: use the consistent format.
Table 4: ND data columns can be removed.
Figure 3: better to mark the places with deposition of brown pigment and loss of luster.

·

Basic reporting

1. This paper has a substantial weak point, in which "summer atrophy" of Akoya pearl oyster is not well defined and validity of diagnostic criteria used in this manuscript (nacre abnormality) is not clear. The characteristics of the disease is described in the introduction part, but no references are given. The authors should have described the disease more in detail with their own data or using references. I would like to recommend the authors give references for the description of the disease, mortality of affected oysters with the detailed description of methods to calculate the mortality. Although I guess that peer-reviewed references are not available as the disease is a newly emerging disease, any information sources including media release by local governments or special/temporal reports from local fisheries experts/organization should be given. I recommend the authors to compare the occurrence of nacre abnormality among much more Akoya oysters with atrophied muscle and healthy oysters and among areas/populations with/without mortality probably caused by the disease to give validity to the diagnostic criteria.

2. Many incorrect or unclear expressions are found in English and scientific wordings. I strongly request the authors to have professional English editing for the revised manuscript.
I am sorry that I cannot give suggestions to improve the expressions, as my native language is not English.

3. Some other basic comments
1) Mortality of oysters at the collection sites should be given in Table 1 to the severity of the diseases in the sites, if available.
2) "average" is not a mathematical word. "mean" should be used in scientific papers.
3) Give the amount of diatoms given to oysters during the experiments.
4) The authors stated that nacre abnormalities was used for diagnosis, as scarification of oysters was needed to know extent of atrophy(line 101-103). However, it is also difficult to examine the nacre abnormalities without killing oysters. It is unclear why the nacre abnormality was used for diagnosis.
5) It is unclear whether oysters were individually separated or the two groups were separated.
6) The results of Test 2 should be given in a table.
7) In Table 4, the lines of "ND" can be deleted.

Experimental design

The experiments are well designed.

Validity of the findings

The authors used student's t-test for comparison of shell score. However, the shell score is obviously non-parametric data, for which non-parametric statistical methods should be used instead of t-test.

Reviewer 3 ·

Basic reporting

No comment.

Experimental design

Although the study provides convincing results on the nature of the etiological agent, further tests could have been carried out in order to infirm/confirm the nature of the agent. For example, PCR tests specific for birnaviruses/Akoya virus (if available), as well as using a virucidal treatment for the infectious hemolymph that would inactivate the agent not allowing infection in healthy oysters. Electron microscopy performed on infected oyster tissues would also provide useful information and strengthen the manuscript.

Validity of the findings

No comment.

Additional comments

I recommend the publication of this manuscript as it provides useful additional knowledge to the field of oyster diseases. Further work will hopefully lead to the identification of this potentially very serious etiological agent.

---

## Round 0.2 · Minor Revisions

We are almost there. The reviewers proposed very few remaining corrections which you should consider.

I look forward to your revised manuscript.

Reviewer 1 ·

Basic reporting

No comment.

Experimental design

No comment.

Validity of the findings

No comment.

Additional comments

The revised manuscript has been much improved based on the experiments of thermal treatment. As for the second major question of the reviewer, a certain metabolic change may have happened by summer atrophy disease, resulting in “increased numbers of migrating cells”. If so, not only histopathological experiments but also biochemical or molecular biological approach will be required. However, this paper contributes to the understanding of pearl summer atrophy disease of pearl oyster even at the present stage, because the pearl oyster aquaculture industries have not established yet any effective strategies against this enormous infectious disease.

Minor comments.
Table 4: death -> the dead?
Table 5: seriese -> series
Table 8: What is “suer atrophy” in the title?

·

Basic reporting

The usage of 'tense' is so unique that it may lead readers to misunderstandings.
They frequently use simple present tense to describe the observations and opinions of limited numbers of previous reports, which have not still been widely recognized as truth.
I think the authors should reconsider the usage of tense in this manuscript.

Experimental design

no comment

Validity of the findings

no comment

---

## Round 0.3 · accepted · Accept

Thank you very much for the thorough revision of your manuscript and the consideration of the reviewer's proposals.